# Livestock Grazing Impact on Species Composition and Richness Understory of the *Pinus cembroides* Zucc. Forest in Northeastern Mexico

**Juan A. Encina-Domínguez** [1], **Eduardo Estrada-Castillón** [2], **Miguel Mellado** [3], **Cristina González-Montelongo** [4] **and José Ramón Arévalo** [4,*]

1 Department of Natural Resources, Autonomous Agrarian University Antonio Narro, Saltillo 25000, Mexico; juan.encinad@uaaan.edu.mx
2 Faculty of Forestry Sciences, Autonomous University of Nuevo León, San Nicolás de los Garza 67700, Mexico; andres.estradacs@uanl.edu.mx
3 Department of Animal Nutrition, Autonomous Agrarian University Antonio Narro, Saltillo 25000, Mexico; miguel.mellado@uaaan.edu.mx
4 Department of Botany, Ecology and Plant Physiology, University of La Laguna, 38206 La Laguna, Spain; cgonzalm@ull.edu.es
* Correspondence: jarevalo@ull.edu.es; Tel.: +34-922318628

**Abstract:** In the pine forests of Mexico, disturbances are primarily due to cattle, horses, goat, and sheep grazing, particularly in communal grazing lands. The most evident disturbances are low tree recruitment, invasive shrubs establishment, species composition changes, and invasion of weeds dispersed mainly by livestock. The Sierra de Zapalinamé is a mountain range and natural protected area of northeast Mexico. We conducted the current study in this area in a forest stand of *Pinus cembroides* excluded from grazing in the last 25 years (1200 ha with pine forest vegetation and mountain chaparral) and another area nearby subjected to livestock grazing. Forest structure (basal area and density), tree species richness, total understory species richness, and understory species composition were analyzed at the control and grazed sites. Our results revealed that grazing has modified the understory species composition and reduced the evenness in the control plots. Therefore, to maintain species diversity and forest structure, we concluded that extensive grazing should be restricted for some areas or the number of animals reduced in zones of high ecological value.

**Keywords:** biodiversity; CCA; conservation; evenness; richness

## 1. Introduction

Livestock grazing is one of the most important activities worldwide [1], requiring correct techniques to maintain species composition and soil conservation in grasslands [2], as these cases can cause remarkable and significant variation in plant species composition [3–5]. In the pine forests of northern Mexico, grasslands disturbance is due to overgrazing by cattle, horses, goats, and sheep, particularly in communal areas [6]. The most evident disturbances are low tree recruitment and the establishment of invasive shrubs and weeds dispersed mainly by livestock.

Mexico is second in the diversity of pinyon pine woodlands after Eurasia [7]. The distribution of Mexican pinyons includes mountainous zones of Arizona, New Mexico, and Texas in the southwest United States of America and from northern and central Mexico down to the state of Puebla [8]. The pinyon pine forests, dominated by *Pinus cembroides* Zucc., occupy extensive areas in the eastern and western Sierra Madre mountains [9]. These drought-tolerant pines develop at altitudes of 1800 to 2800 m, on dry soils and rocky slopes of mountains and hills with poorly developed soils, in temperate dry to temperate sub-humid zones. They are common in transition zones between xeric and relatively mesic forest communities at higher elevations [10].

The Sierra de Zapalinamé is a mountainous protected area enacted by the state of Coahuila, Mexico [11]. Here, the pinyon pine forest grows on low slopes with slight slants and inter-mountain valleys with deep soils in temperate sites at altitudes from 2150 to 2650 m [12]. Little is known about the effects of heavy grazing in this type of vegetation on forest stands and associated vegetation, even though forest grazing is a common management practice applied worldwide [13]. Thus, it was considered pertinent to shed light on vegetation diversity in pinyon pine forest to elaborate guides for pinyon pine forest richness conservation in northeastern Mexico. Also, it would be valuable to improve the soil quality, restore vegetation and recover wildlife in an overgrazed pinyon pine forest.

The main objective of this study was to analyze the impact of livestock grazing exclusion for 25 years on species composition and soil nutrients in a pinyon pine forest. We hypothesized that *Pinus cembroides* forests excluded from grazing for a prolonged period have a positive impact on species richness and species composition and that the effect on nutrient composition will also be significant. Therefore, determining species composition when combining pines and cattle can be valuable for managers in the decision-making procedure for *Pinus cembroides* forests.

## 2. Materials and Methods

### 2.1. Study Site

The Sierra of Zapalinamé is located in northeastern Mexico, and it has an area of 45,000 ha. It is located south of the city of Saltillo, between 25°15′00″ N and 25°25′58″ N and between 100°47′14″ W and 101°05′03″ W (Figure 1) and belongs to the Gran Sierra Plegada physiographic subprovince. The elevation ranges from 1590 m in the foothills to 3140 m in the highest mountain, with inter-mountain valleys averaging 2200 m a.s.l. Rocks of the area are sedimentary, belonging to the Jurassic and Cretaceous periods; limestone covers 43% of the area, while 17% is sandstone and conglomerates [14]. Alluvial soils occupy 30% of the area with variable depth and are mainly found in the plains with alluvial fans at the mountain base. Soils in the valleys are deep. There are also smaller areas of calcium and phaeozem calcaric xerosols.

The prevailing climate of the study area is the dry type (BSkw), while the upper parts of the mountain have a temperate type (C(w0)) following Köppen's classification. The average annual temperature is 16.9 °C, and the mean annual rainfall is 498 mm [14]. Rains are convective and occur mainly in the warmest months of the year. Different plant communities have been recorded for this area, including rosetophyllous scrub, pine forest, fir forest, oak forest, and montane chaparral. In the protected area, pinyon pine forest occupies 12.54% of the area surrounded by a xeric scrubland (9.55% of the protected area) [15]. Pinyon pine forests are distributed mainly in the Cuauhtémoc and Sierra Hermosa canyons; *Pinus cembroides* grow scattered among *Juniperus flaccida* Schltdl. and *J. deppeana* Steud. On the branches of pine trees, the epiphytic *Tillandsia recurvata* (L.) L. is common [16]. In more conserved areas, the herb layer is dominated by the grasses *Piptochaetium fimbriatum* (Kunth) Hitchc. and *Bouteloua dactyloides* (Nutt.) Columbus. In areas with intense perturbation by cattle grazing, *Asphodelus fistulosus* L. and *Gymnosperma glutinosum* Less. are abundant [15]. In this forest, the greatest perturbation is caused by human intervention, through land-use change to establish agricultural areas, causing fragmentation and surface area reduction.

This study was conducted in a forest stand excluded from grazing for 25 years on the San José del Anhelo private property (1200 ha with pinyon pine forest and mountain chaparral). This area was used to establish the grazing-excluded plots. This exclusion management technique was intended to improve soil quality, restore vegetation and recover wildlife. Outside the private property, in the same potential vegetation stand, we located the grazed plots (Ejido Cuauhtémoc, 270 ha), where extensive grazing takes place with 104 cows, 18 donkeys, 28 horses, 683 goats, and 84 sheep (this information has remained relatively constant in the studied area for the last 20 years according to personal commu-

nication from the protected area managers). Owners of this site selectively log pines for house and fences construction, Christmas trees, and harvest of pinyon nuts. These activities have become the main economic activity of the surrounding settlements. Therefore, we avoided the areas with intensive management or use in our study.

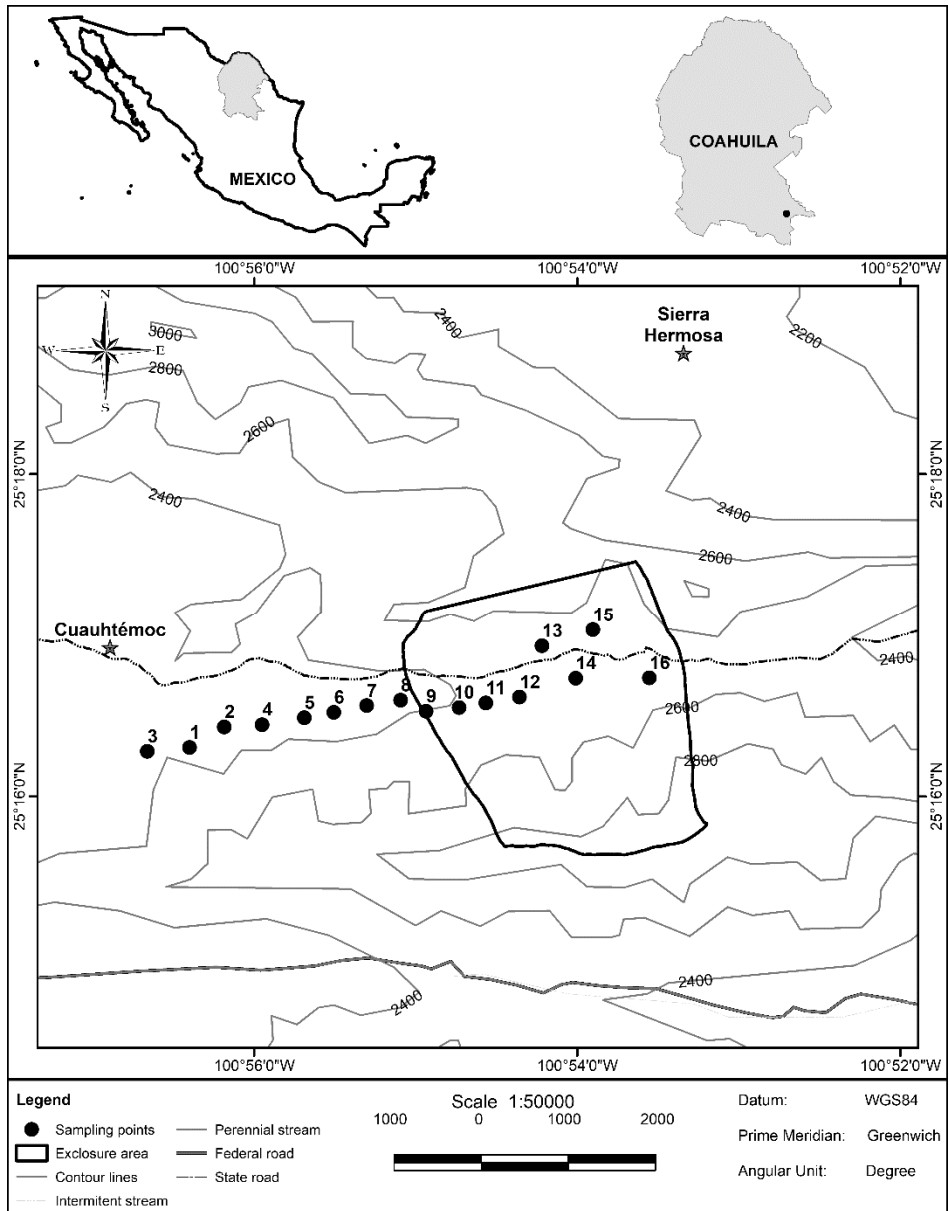

**Figure 1.** Location of the Protected Natural Area Sierra of Zapalinamé in northern Mexico (25° N). The grazing-excluded area is indicated (surrounded by a grey line), as well as the location of the plots.

## 2.2. Sampling Design

In August 2017, we systematically located 16 (30 × 30 m) rectangular plots in a stand of *Pinus cembroides* in the Sierra of Zapalinamé natural protected area. We established eight plots in the grazed area (grazed plots) and eight in the grazing-excluded area (control plots). Plots were located along a transect in the center of the stand at a distance of 100 m (avoiding trails or human disturbances). In each plot, we measured land altitude and slope and estimated the canopy cover of the stand using a convex spherical crown densitometer [17]. We visually estimated the percentage cover of rock, bare soil, and litter cover in each plot. Grass cover (only grasses excluding forbs) and understory woody species cover also were visually estimated. We considered trees to be individuals whose

stems were $\geq$2.5 cm DBH (diameter at breast height). We measure DBH for all trees alive in the 30 $\times$ 30 m plots to estimate basal area and density to ha. Previous studies recommended these classifications following the physiognomy of these species [18]. This category will be considered as part of the canopy for its description.

We identified all herbs and shrubs in a concentric plot of 10 $\times$ 10 m. Cover for all the species on plot surfaces was estimated and recorded on a scale of 1 to 9 (cover classes: 1: traces; 2: >1% of cover in the plot; 3: 1%–2%; 4: 2%–5%; 5: 5%–10%; 6: 10%–25%; 7: 25%–50%; 8: 50%–75%; 9: >75%). Taxonomic identities of collected plant specimens were determined and vouchers were deposited at the ANSM herbarium. For species names, we followed the checklist of vascular plants of the Sierra of Zapalinamé [12]. Plot position and elevation were measured using a global positioning system (GPS; Etrex, Garmin Ltd., Olathe, KS, USA).

We took four soil samples (0 to 10 cm in depth in each corner of the plots). These were mixed, dried, and sifted through a 2-mm sieve; debris and stones were eliminated. Organic matter content was determined by the Walkley and Black method [19], and pH was measured in a soil-to-water ratio of 1:5 extract. Soil total nitrogen (the Kjeldahl method), phosphorus Olsen (0.5 M NaHCO$_3$ with a soil/solution ratio of 1:20 [20]), K, Na, Mg, Ca, and electrical conductivity were determined. After adding lanthanum to a solution, Ca and Mg contents were determined by the ASS Manual (Wollongbar Agricultural Institute, Australia). Soil K and Na concentrations were determined after extraction with the AL-method (open vessel extraction with ammonium lactate and acetic acid). We also calculated Cation Exchange Capacity. The qualitative levels baseline for nutrients obtained, such as N, P, K, and organic matter, were according to SEMARNAT (2000) [21] and Fernandez-Linares et al. (2006) [22].

### 2.3. Statistical Analysis

A one-way distance-based permutational t-statistic [23] was performed for comparison between grazed and control plots (as factors) for species richness, Smith and Wilson evenness [24], basal area (m$^2$/ha), and tree density (individuals/ha). The analyses were based on the Bray–Curtis distance of the raw data, with $p$-values < 0.05 obtained with 9999 permutations and a Monte Carlo correction where necessary. Primer 6 and Permanova+ (PRIMER-E Ltd., Plymouth, UK) were used to perform all PERMANOVA statistical procedures. We used the same comparison analysis for the following soil nutrient and environmental variables: pH, EC (exchangeable cations, dS/m), Polsen (Phosphorus Olsen extraction in ppm), organic matter (% OM), available cations in meq/100 (Na, K, Ca, Mg), cation exchange capacity (CEC) and total nitrogen (% TN), soil cover, grass cover, woody species cover, litter cover estimated in percentages and canopy cover.

As a technique of direct gradient analysis, we used partial Canonical Correspondence Analysis (CCA; [25]) in CANOCO 5.1. [26] to examine how species composition changed over the different plots as a function of the environmental characteristics included in the analysis. In the environmental matrix, we used the following variables: pH, EC (exchangeable cations, dS/m), Polsen (Phosphorus Olsen extraction in ppm), % OM, available cations in meq/100 (Na, K, Ca, Mg), CEC, a and %TN. Additionally, we included soil cover, grass cover, woody species cover, litter cover estimated in percentages, and canopy cover. As a biotic matrix, we used the total species composition based on the cover of the 10 $\times$ 10 m plots. We selected the three most informative environmental variables, applying a forward selection procedure to remove the variables that did not explain a significant portion of the variability reported by the analysis when performing the axes (Monte Carlo permutation test with 9999 interactions for $p < 0.05$). Axes I and II are graphically displayed with the selected environmental variables and plots enclosed in a different polygon for grazed vs. control. Species are presented separately in the same bio-dimensional space of CCA axes I and II.

An MRPP (Multi-response Permutation Procedure) was used to determine changes in species composition between grazed and control plots with a matrix base in cover. The

Bray-Curtis distance was used for this analysis [27]. For the same data matrix, a Species Indicator Analysis was used to determine the significant representative species in each group [28]. The analyses were carried out in the *vegan* R package [29].

## 3. Results

The altitude of the plots ranged between 2350–2500 and as for slope and canopy cover, differences were not significant for control vs. grazed plots. Grass cover was higher (PseudoF$_{1,14}$ = 19.87, $p < 0.01$) in the control plots, while woody species cover, soil, and litter were higher in the grazed plots (PseudoF$_{1,14}$ = 8.20, PseudoF$_{1,14}$ = 5.45 and PseudoF$_{1,14}$ = 11.18 respectively, with a $p < 0.05$ for the first two and $p < 0.01$ for the last one). The rest of the variables, including the nutrient content, did not reveal significant differences (Table 1).

We found 12 tree species in the study site, 6 in the grazed plots, and 11 in the control plots. *Fraxinus greggii* A. Gray was not present in the control plots, while *Arbutus xalapensis* Kunth, *Arctostaphylos pungens* Kunth, *Pinus arizonica* Engelm., *Rhus virens* Lindh. *ex* A. Gray and *Yucca carnerosana* (Trel.) McKelvey were not present in the grazed plots. *Pinus cembroides*, *Juniperus deppeana*, *J. flaccida*, *J. coahuilensis* (Martínez) Gaussen *ex* R.P. Adams, *Quercus saltillensis* Trel. and *Q. microphylla* Née were present in both study sites.

*Pinus cembroides* was the dominant species with higher values of basal area in grazed plots (85.6%) than in control plots (72.0%). Tree species richness differed significantly between sites with 3.6 ± 1.6 (mean ± standard deviation) for the eight plots in grazed plots and 6.0 ± 1.3 (mean ± standard deviation) in control plots (PseudoF$_{1,14}$ = 10.51, $p < 0.05$). There were no significant differences for density and basal area (PseudoF$_{1,14}$ = 2.83 and PseudoF$_{1,14}$ = 1.50, $p$ = n.s. respectively) between sites. More detailed information about the canopy can be found in Arévalo et al. [16].

**Table 1.** General abiotic information of experimental plots and soil nutrients and characteristics of experimental plots. The values' average (Avg) and standard deviations (Std) for grazed and control plots are included.

| Plots | Treatment [1] | Alt (m) | Slope | Grass * | Woody * | Rock | % Cover Soil * | Litter * | Canopy |
|---|---|---|---|---|---|---|---|---|---|
| PG1 | Grazed | 2351 | 10 | 25 | 25 | 3 | 1 | 60 | 53.6 |
| PG2 | Grazed | 2346 | 11 | 60 | 5 | 2 | 2 | 20 | 47.2 |
| PG3 | Grazed | 2342 | 39 | 20 | 10 | 3 | 10 | 10 | 56.6 |
| PG4 | Grazed | 2356 | 28 | 30 | 15 | 15 | 20 | 10 | 52.8 |
| PG5 | Grazed | 2372 | 10 | 40 | 10 | 2 | 30 | 15 | 55.6 |
| PG6 | Grazed | 2372 | 10 | 55 | 15 | 1 | 30 | 20 | 49.8 |
| PG7 | Grazed | 2379 | 9 | 50 | 10 | 1 | 30 | 20 | 52 |
| PG8 | Grazed | 2394 | 10 | 45 | 10 | 1 | 45 | 15 | 55.2 |
| **Average** | | 2364.0 | 15.9 | 40.6 | 12.5 | 3.5 | 21.0 | 21.3 | 52.9 |
| **SD** | | 18.1 | 11.3 | 14.5 | 6.0 | 4.7 | 15.6 | 16.2 | 3.2 |
| PC2 | Control | 2419 | 10 | 80 | 5 | 1 | 2 | 2 | 55.4 |
| PC3 | Control | 2436 | 10 | 70 | 10 | 2 | 5 | 3 | 57.2 |
| PC4 | Control | 2450 | 12 | 70 | 10 | 1 | 2 | 7 | 43.6 |
| PC5 | Control | 2468 | 10 | 85 | 5 | 1 | 2 | 10 | 52.2 |
| PC6 | Control | 2466 | 11 | 85 | 7 | 1 | 1 | 2 | 36.6 |
| PC7 | Control | 2498 | 11 | 65 | 5 | 3 | 15 | 7 | 53.6 |
| PC8 | Control | 2501 | 11 | 70 | 5 | 1 | 1 | 10 | 42 |
| **Average** | | 2458.4 | 10.8 | 75.6 | 6.5 | 1.4 | 4.0 | 5.8 | 49.8 |
| **SD** | | 30.5 | 0.7 | 7.8 | 2.3 | 0.7 | 4.7 | 3.3 | 7.9 |

**Table 1.** *Cont.*

| Plots | pH | % OM | % TN | ppm P Ols | µS/cm EC | Ca | Mg | meq/100 g Na | K | CEC |
|-------|-----|------|------|-----------|----------|------|------|--------------|------|------|
| PG1 | 7.5 | 15.5 | 0.63 | 21.65 | 1048 | 4.4 | 12.4 | 10,463.2 | 0.2 | 39 |
| PG2 | 7.2 | 13.8 | 0.64 | 20.13 | 1223 | 12 | 16 | 12,202 | 0.2 | 12 |
| PG3 | 5.2 | 14.5 | 0.81 | 25.36 | 1387 | 12 | 6 | 13,852 | 0.28 | 29 |
| PG4 | 7.4 | 17.5 | 0.75 | 11.36 | 1471 | 12 | 8 | 14,690 | 0.18 | 30 |
| PG5 | 4.0 | 14.5 | 0.47 | 8.35 | 1580 | 10 | 10 | 15,780 | 0.31 | 25 |
| PG6 | 7.6 | 6.9 | 0.4 | 17.37 | 1262 | 20 | 4.8 | 12,595.2 | 0.22 | 21 |
| PG7 | 6.8 | 7.8 | 0.27 | 3.01 | 916 | 8.4 | 11.6 | 9140 | 0.26 | 20 |
| PG8 | 2.3 | 6.2 | 0.26 | 3.28 | 374 | 12 | 20 | 3708 | 0.22 | 30 |
| **Avg** | 6.0 | 12.1 | 0.5 | 13.8 | 1157.6 | 11.4 | 11.1 | 11,553.8 | 0.2 | 25.8 |
| **Std** | 2.0 | 4.4 | 0.2 | 8.5 | 383.4 | 4.4 | 5.1 | 3837.0 | 0.0 | 8.2 |
| PC1 | 7.3 | 17.8 | 0.58 | 11.36 | 1238 | 8.4 | 12 | 12,359.6 | 0.22 | 30 |
| PC2 | 4.0 | 7.4 | 0.27 | 11.64 | 971 | 10.8 | 9.2 | 9690 | 0.2 | 24 |
| PC3 | 7.3 | 18.2 | 0.5 | 6.71 | 1282 | 14 | 10 | 12,796 | 0.41 | 32 |
| PC4 | 7.7 | 7.8 | 0.31 | 3.56 | 1022 | 8.8 | 12 | 10,199.2 | 0.17 | 30 |
| PC5 | 7.5 | 9.1 | 0.46 | 10.54 | 970 | 8 | 4 | 9688 | 0.27 | 25 |
| PC6 | 7.6 | 24.5 | 0.77 | 17.65 | 1366 | 14 | 6 | 13,640 | 0.17 | 14 |
| PC7 | 7.3 | 14.5 | 0.77 | 17.65 | 1027 | 10 | 10.4 | 10,249.6 | 0.23 | 25 |
| PC8 | 7.0 | 17.5 | 0.71 | 18.89 | 1365 | 15.2 | 16.8 | 13,618 | 0.27 | 12 |
| **Avg** | 6.9 | 14.6 | 0.5 | 12.3 | 1155.1 | 11.2 | 10.1 | 11,530.1 | 0.2 | 24.0 |
| **Std** | 1.2 | 6.1 | 0.2 | 5.5 | 174.8 | 2.9 | 3.9 | 1744.0 | 0.1 | 7.4 |

(*) Indicate significant differences between grazed vs. control plots for a *p* < 0.05. (1) "Grazed" for plots where grazing has been active for the last 25 years, and "control" for non-grazed plots for all this period. OM: Organic matter; TN: Total nitrogen; P Ols: Phosphorus Olsen; EC: Electric Conductivity; CEC: Cation exchange capacity.

A total richness of 163 understory species was found in the pinyon pine forest; the families with the most plant species were Asteraceae with 45 species, Poaceae 18, and Fabaceae 8. Herbs were the most common biological form, with 119 species. In addition, we recorded 55 species in the control plots, which did not occur in the grazed plots, while 34 species were present only in grazed plots (Appendix A).

Concerning species richness, non-significant differences existed between experimental sites. At the same time, Smith and Wilson Evenness revealed significant differences, with higher values in the grazed than the control plots (PseudoF$_{1,14}$ = 1.47, *p* = n.s. and PseudoF$_{1,14}$ = 5.20, *p* < 0.05 respectively; Figure 2).

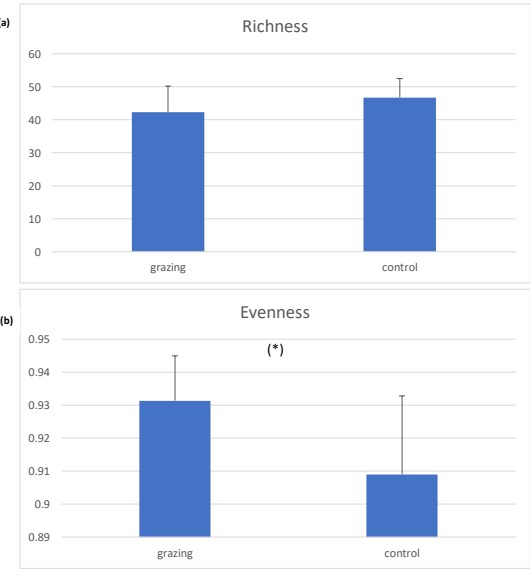

**Figure 2.** Mean values and standard deviations for (**a**) total species richness and (**b**) evenness for grazing and control plots. (*) For significant differences (*p* < 0.05).

The CCA analysis revealed that of all the physical parameters analyzed, only three were significant for the distribution of the species composition: grass cover, canopy cover, and soil total nitrogen. First, the CCA axis discriminated grazed vs. control plots, with grass cover more representative in the control plots and canopy cover more representative in the grazed plots. However, axis 2 was related to %TN with higher values in the grazed plots (Figure 3). In the case of the species, a gradient of variation in species composition was revealed from control to grazed plots.

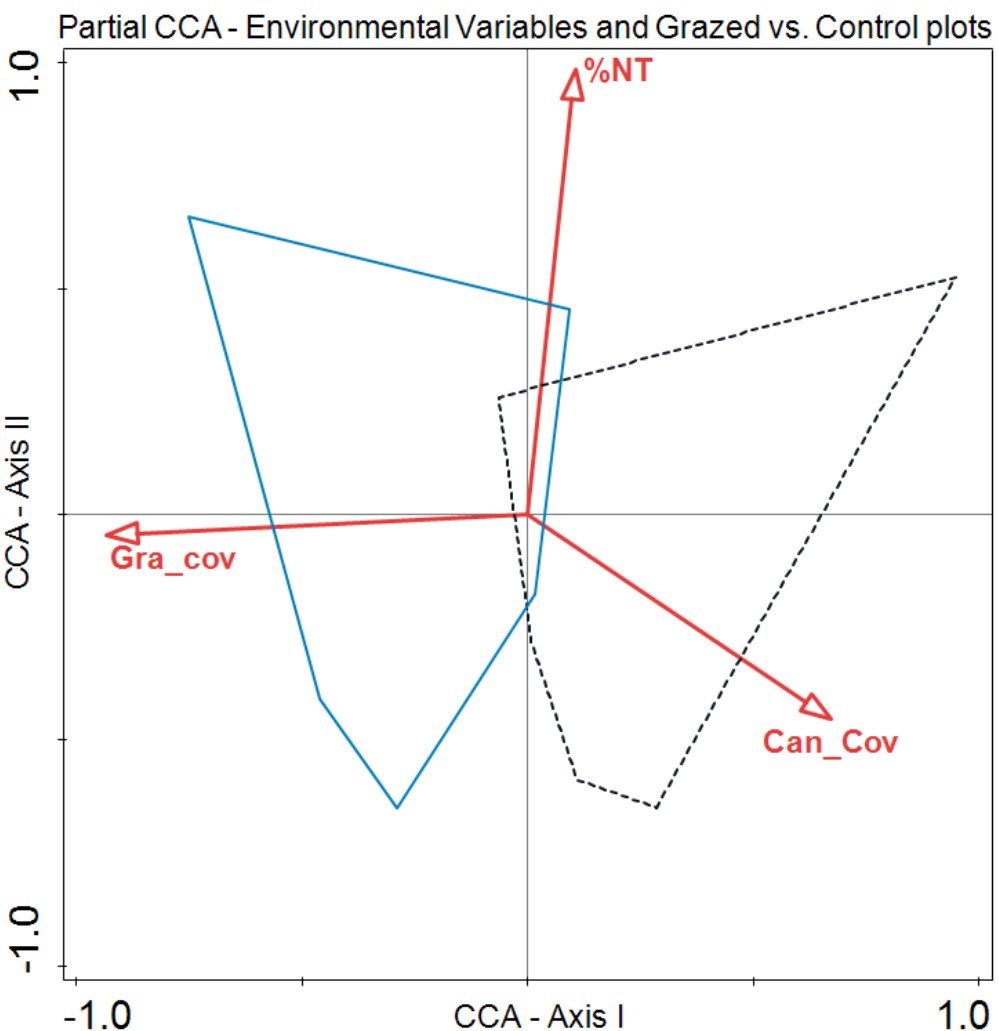

**Figure 3.** Canonical correspondence analysis using five selected environmental variables: grass cover (Gra-cov), canopy (Can_Cov), and total nitrogen (%NT). It also includes envelopes enclosing the two groups of plots (control-solid line) and grazed plots (dashed line). Eigenvalues axis I: 0.28; axis II: 0.20; both axes cumulative percentages explained for the species composition 19.5% and both axes cumulative percentages variance explained for the species-environmental relationship: 71.5%.

Differences between treatments based on species cover were significant (MRPP) with a T = −7.102 and group probability correction of A = 0.087 for a $p < 0.01$.

Finally, the ISA base in 1000 permutations revealed that the indicator species for the grazed plots were *Bouteloua dactyloides*, *Crusea diversifolia,* and *Dichondra brachypoda*, while *Dalea radicans*, *Muhlenbergia rigida* (Kunth) Kunth, *Arbutus xalapensis*, *Hedeoma costata* A. Gray, *Juniperus deppeana*, and *Malaxis brachystachys* (Lindl.) Rchb. f. were indicators for the control plots ($p < 0.01$ for all species). These indicator species were represented in the CCA biplot in bold letters (Figure 4).

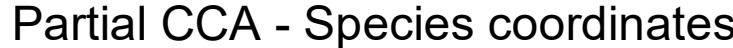

**Figure 4.** Canonical Correspondence analysis axes I and II with the species coordinates. Species names in bold are indicator species for grazing or control plots (negative scores with respect to axis I are for control plots while positive scores are for grazed plots). Species names use the three first letters of the genus, followed by the first three letters of the specific epithet from Appendix A.

## 4. Discussion

Some environmental characteristics markedly differed between sites, such as grass cover, soil cover, litter and woody species cover, and characteristics related to grazing intensity [30]. However, differences in nutrient content were not significant. The intensity of herbivores' effect on plant communities varies along environmental gradients or vegetation stands [31], even being insignificant or null at low grazing intensities [32–34].

For forest structure (basal area and density), differences were not significant, although the tree species richness was higher in control plots than in grazed plots, indicating an adverse grazing impact on some species such as *Arbutus xalapensis* or *Yucca carnerosana* [16]. These species are highly palatable for goats and cattle [35,36]. Some studies have demonstrated that grazing impacts species richness [37,38].

Although it is dependent on a spatial scale and highly related to climate variability [39,40] and resource availability [38,41]. In the present study, this impact was significant for tree species, although basal area and density of trees did not differ, suggesting that grazing exclusion did not promote changes in forest structure for that period.

Regarding total species number, we found non-significant differences. However, evenness was higher in grazed plots. This can be seen with the reduction through grazing of dominant species and the larger values of grass cover in the control plots. Several studies have found higher evenness in grazed than ungrazed treatments [40,42,43]. This was also reflected in the current study. However, some studies measuring grazing effects on plant species composition and species richness have traditionally been inconsistent and conflicting in their results, lacking a general model that predicts the response of grazing intensity or abandonment [37,44], and the lack of consistent results has been attributed to high factors variability such as the evolutionary history of grazing, productivity gradients or grazing intensity [33].

The ordination analyses revealed that grazed plots vs. control plots are discriminated based on the total species cover, and only three environmental variables significantly explained the species composition. They were grass cover, canopy cover, and total nitrogen. Nitrogen was not an important discriminant variable among sites and was more related to particular conditions of the soil. However, grass cover and canopy cover were important variables explaining species composition in control and grazed plots. The reduction of grass cover is a typical result [45], but grazing does not typically impact canopy cover. Canopy cover and grass cover are inversely related [46], but some studies have revealed a low relationship between canopy cover and grazing [47].

In control plots, two terrestrial orchids grow: *Malaxis brachystachys* and *Goodyera oblongifolia* Raf. Moreover, species like *Dahlia tubulata* P.D. Sørensen, *Geranium semannii*, *Gibasis geniculata* (Jacq.) Rohweder and *Salvia regla* Cav. Are frequent but did not occur in the grazed plots. Both species groups prefer to grow in mesic forest conditions and undisturbed sites. These species are common in oak forests in this region [18], growing in humid canyons.

The grasses with high coverage in the herbaceous stratum in the control plots were *Muhlenbergia rigida* and *Piptochaetium fimbriatum*, both tufted grasses that grow up to 1.0 m tall. On the other hand, in the grazed plots, *Bouteloua dactyloides* was the dominant species. This is a cespitose, stoloniferous short grass that forms a short carpet up to 10 cm high [48]. According to Encina-Domínguez et al. (2019) [15], it is a common grass in the Zapalinamé mountain range, growing in remnant semi-desert grasslands located in valleys with deep soils that support intense cattle and horse grazing.

In grazed plots, annual weeds [49], such as *Bidens odorata*, *Dyssodia papposa* (Vent.) Hitchc., *D. pinnata* (Cav.) B.L. Rob., *Euphorbia dentata* Michx., *Lepidium virginicum* L. and *Viguiera dentata* (Cav.) Spreng also grow. These are dispersed by grazing livestock from croplands adjacent to the forest. It is noteworthy that *Prunus cercocarpifolia* Villarreal was found only in grazed forests, a shrubby rhizomatous species endemic to this mountain range [12]. In these areas, the cacti *Echinocereus knippelianus* Liebner, in conservation status by the Mexican government, also grows [50].

We only found one exotic species on these analyzed pastures. In grazed plots, a perennial weed *Asphodelus fistulosus* L. an exotic species from Eurasia grows with scattered individuals. In the mountain range, it is common along the roads, abandoned agricultural fields, and overgrazed areas. In this area, with a long dry season and a very cold winter, both are two strong environmental filters that may limit the establishment of many native ruderal and exotic species [51], which can explain its low number.

Species composition was well discriminated, with some species as the control or grazed plots indicators. Shrubs and trees were indicators on control plots, together with some other herbs, while herbs only are indicators of grazed plots. Livestock grazing reduced highly palatable shrubs, particularly by goats. In general, there was a species turn-over from the grazed to control plots, being more similar to the climax vegetation of these forests. Future research directions may also be highlighted.

## 5. Conclusions

Uncontrolled livestock grazing has modified the species composition in the *Pinus cembroides* forest. The number of trees has been affected negatively, reducing the number of species. Other studies have also revealed an adverse effect of cattle grazing on species richness and plants [52–54].

We conclude that extensive grazing carried out for decades in the *Pinus cembroides* stand of the Sierra de Zapalinamé should be restricted or the number of animals reduced in zones of high ecological value, to maintain diversity and forest structure. Livestock grazing is a necessary activity for the economy of farmers in communal lands because of the meat products obtained from cattle, goats, and sheep in the local area. We suggest the application of controlled grazing pressure for some areas of particular conservation interest to restore mature, persistent *P. cembroides* forest to a more historical condition.

**Author Contributions:** Conceptualization, J.A.E.-D. and J.R.A.; methodology, J.A.E.-D., E.E.-C. and C.G.-M.; data curation, J.A.E.-D., E.E.-C. and C.G.-M.; supervision, J.A.E.-D., J.R.A. and M.M.; writing—review and editing, J.A.E.-D. and J.R.A.; project administration, J.A.E.-D. All authors have read and agreed to the published version of the manuscript.

**Funding:** This research received no external funding.

**Data Availability Statement:** Not applicable.

**Acknowledgments:** We wish to thank the staff of the Zapaliname protected area for supporting this research, especially Sergio C Marines Gómez. We also thank Arturo Cruz-Anaya, Leticia Jiménez and Rocío Martínez for their assistance during field data collection. Thank you to Jerome Scorer for proofreading and amending the English version of this paper. Many thanks to the Universidad de La Laguna in Tenerife, Spain, for their invaluable support during the preparation of this paper.

**Conflicts of Interest:** The authors declare no conflict of interest.

## Appendix A. Species Family, Scientific Name and Life Form Found in This Study

| Family | Scientific Name | Species Abbreviations | Life Form |
|---|---|---|---|
| Acanthaceae | *Dyschoriste linearis* (Torr. & A. Gray) Kuntze | *Dys lin* | Herb |
| | *Elytraria imbricata* (Vahl) Pers. | *Ely imb* | Herb |
| Amaranthaceae | *Chenopodium foetidum* Lam. | *Che foe* | Herb |
| Anacardiaceae | *Rhus aromatica* Aiton | *Rhu aro* | Shrub |
| | *Rhus virens* Lindh. *ex* A. Gray | *Rhu vir* | Shrub |
| Apiaceae | *Donnellsmithia ternata* (S. Watson) Mathias & Constance | *Don ter* | Herb |
| Asparagaceae | *Agave gentryi* B. Ullrich | *Aga gen* | Shrub |
| | *Dasylirion cedrosanum* Trel. | *Das ced* | Shrub |

| Family | Scientific Name | Species Abbreviations | Life Form |
|---|---|---|---|
| | *Nolina cespitifera* Trel. | *Nol ces* | Shrub |
| | *Yucca carnerosana* (Trel.) McKelvey | *Yuc car* | Shrub |
| Asphodelaceae | *Asphodelus fistulous* L. | *Asp fis* | Herb |
| Asteraceae | *Acourtia wrightii* (A. Gray) Reveal & R.M. King | *Aco wri* | Herb |
| | *Ageratina calophylla* (Greene) Molinari & Mayta | *Age cal* | Herb |
| | *Ageratina saltillensis* (B.L. Rob.) R.M. King & H. Rob. | *Age sal* | Shrub |
| | *Ageratina scorodonioides* (A. Gray) R.M. King & H. Rob. | *Age sco* | Herb |
| | *Ageratum corymbosum* Zuccagni | *Age cor* | Herb |
| | *Artemisia ludoviciana* Nutt. | *Art lud* | Herb |
| | *Aztecaster matudae* (Rzed.) G.L. Nesom | *Azt mat* | Shrub |
| | *Baccharis potosina* A. Gray | *Bac pot* | Shrub |
| | *Bidens pilosa* L. | *Bid pil* | Herb |
| | *Brickellia eupatorioides* (L.) Shinners | *Bri eup* | Herb |
| | *Brickellia grandiflora* (Hook.) Nutt. | *Bri gra* | Herb |
| | *Brickellia lemmonii* A. Gray | *Bri lem* | Herb |
| | *Brickellia veronicifolia* (Kunth) A. Gray | *Bri ver* | Shrub |
| | *Chaetopappa bellioides* (A. Gray) Shinners | *Chae bel* | Herb |
| | *Chaptalia nutans* (L.) Pol. | *Cha nut* | Herb |
| | *Chrysactinia mexicana* A. Gray | *Chr mex* | Shrub |
| | *Dahlia tubulata* P.D. Sørensen | *Dah tub* | Herb |
| | *Dyssodia papposa* (Vent.) Hitchc. | *Dys pap* | Herb |
| | *Dyssodia pinnata* (Cav.) B.L. Rob. | *Dys pin* | Herb |
| | *Erigeron pubescens* Kunth | *Eri pub* | Herb |
| | *Gymnosperma glutinosum* (Spreng.) Less. | *Gym glu* | Shrub |
| | *Helianthella mexicana* A. Gray | *Hel mex* | Herb |
| | *Heterosperma pinnatum* Cav. | *Het pin* | Herb |
| | *Hieracium crepidispermum* Fr. | *Hie cre* | Herb |
| | *Lactuca graminifolia* Michx. | *Lac gram* | Herb |
| | *Pseudognaphalium roseum* (Kunth) Anderb. | *Pse ros* | Herb |
| | *Pseudognaphalium semiamplexicaule* (DC.) Anderb. | *Pse sem* | Herb |
| | *Sanvitalia angustifolia* Engelm. *ex* A. Gray | *San ang* | Herb |
| | *Solidago hintoniorum* G.L. Nesom | *Sol hin* | Herb |
| | *Stevia micrantha* Lag. | *Ste mic* | Herb |
| | *Stevia ovata* Willd. | *Ste ova* | Herb |
| | *Stevia porphyrea* McVaugh | *Ste por* | Herb |
| | *Stevia salicifolia* Cav. | *Ste sal* | Shrub |
| | *Stevia serrata* Cav. | *Ste ser* | Herb |
| | *Stevia tomentosa* Kunth | *Ste tom* | Herb |
| | *Tagetes lucida* Cav. | *Tag luc* | Herb |

| Family | Scientific Name | Species Abbreviations | Life Form |
|---|---|---|---|
| | *Tetraneuris scaposa* (DC.) Greene | *Tet sca* | Herb |
| | *Thymophylla pentachaeta* (DC.) Small | *Thy pen* | Herb |
| | *Verbesina coahuilensis* A. Gray *ex* S. Watson | *Ver coa* | Herb |
| | *Verbesina hypomalaca* A. Gray *ex* S. Watson | *Ver hyp* | Herb |
| | *Verbesina longipes* Hemsl. | *Ver lon* | Herb |
| | *Vernonia greggii* A. Gray | *Ver gre* | Herb |
| | *Viguiera dentata* (Cav.) Spreng. | *Vig den* | Herb |
| | *Viguiera greggii* (A. Gray) S.F. Blake | *Vig gre* | Shrub |
| | *Zaluzania megacephala* Sch. Bip. | *Zal meg* | Herb |
| Berberidaceae | *Alloberberis eutriphylla* (Fedde) C.C.Yu & K.F.Chung | *All eut* | Shrub |
| Boraginaceae | *Nama hispida* A. Gray | *Nam his* | Herb |
| Brassicaceae | *Lepidium virginicum* L. | *Lep vir* | Herb |
| Bromeliaceae | *Tillandsia recurvata* (L.) L. | *Til rec* | Epiphytic |
| Cactaceae | *Coryphantha hintoniorum* Dicht & A. Lüthy | *Cor hin* | Cacti |
| | *Echinocereus knippelianus* Liebner | *Ech kni* | Cacti |
| | *Opuntia engelmannii* Salm-Dyck | *Opu eng* | Cacti |
| Campanulaceae | *Lobelia ehrenbergii* Vatke | *Lob ehr* | Herb |
| Caprifoliaceae | *Lonicera pilosa* (Kunth) Spreng. | *Lon pil* | Vine |
| Caryophyllaceae | *Arenaria lycopodioides* Willd. *ex* D.F.K. Schltdl. | *Are lyc* | Herb |
| | *Drymaria glandulosa* Bartl. | *Dry gla* | Herb |
| | *Paronychia mexicana* Hemsl. | *Par mex* | Herb |
| Commelinaceae | *Gibasis geniculata* (Jacq.) Rohweder | *Gib gen* | Herb |
| | *Gibasis karwinskyana* (Schult. f.) Rohweder | *Gib kar* | Herb |
| Convolvulaceae | *Dichondra argentea* Humb. & Bonpl. *ex* Willd. | *Dic arg* | Herb |
| | *Dichondra brachypoda* Wooton & Standl. | *Dic bra* | Herb |
| | *Ipomoea costellata* Torr. | *Ipo cos* | Herb |
| Cupressaceae | *Juniperus coahuilensis* (Martínez) Gaussen | *Jun coa* | Shrub |
| | *Juniperus deppeana* Steud. | *Jun dep* | Tree |
| | *Juniperus flaccida* Schltdl. | *Jun fla* | Tree |
| Cyperaceae | *Carex schiedeana* Kunze | *Car sch* | Herb |
| Ericaceae | *Arbutus xalapensis* Kunth | *Arb xal* | Tree |
| | *Arctostaphylos pungens* Kunth | *Arc pun* | Shrub |
| Euphorbiaceae | *Euphorbia brachycera* Engelm. | *Eup bra* | Herb |
| | *Euphorbia dentata* Michx. | *Eup den* | Herb |
| | *Euphorbia exstipulata* Engelm. | *Eup exs* | Herb |
| | *Euphorbia macropus* (Klotzsch & Garcke) Boiss. | *Eup mac* | Herb |
| | *Euphorbia serrula* Engelm. | *Eup ser* | Herb |
| | *Evolvulus sericeus* Sw. | *Evo ser* | Herb |
| | *Tragia ramosa* Torr. | *Tra ram* | Herb |
| Fabaceae | *Astragalus sanguineus* Rydb. | *Ast san* | Herb |

| Family | Scientific Name | Species Abbreviations | Life Form |
|---|---|---|---|
| | *Cologania angustifolia* Kunth | *Col ang* | Herb |
| | *Cologania pallida* Rose | *Col pal* | Herb |
| | *Dalea bicolor* Humb. & Bonpl. *ex* Willd. | *Dal bic* | Shrub |
| | *Dalea capitata* S. Watson | *Dal cap* | Shrub |
| | *Dalea radicans* S. Watson | *Dal rad* | Shrub |
| | *Mimosa aculeaticarpa* Ortega | *Mim acu* | Shrub |
| | *Senna demissa* (Rose) H.S. Irwin & Barneby | *Sen dem* | Herb |
| Fagaceae | *Quercus microphylla* Née | *Que mic* | Shrub |
| | *Quercus pringlei* Seemen | *Que pri* | Shrub |
| | *Quercus saltillensis* Trel. | *Que sal* | Shrub |
| Geraniaceae | *Geranium seemannii* Peyr. | *Ger sem* | Herb |
| Hydrangeaceae | *Philadelphus microphyllus* A. Gray | *Phi mic* | Shrub |
| Lamiaceae | *Hedeoma costata* A. Gray | *Hed cos* | Herb |
| | *Salvia glechomifolia* Kunth | *Sal gle* | Herb |
| | *Salvia greggii* A. Gray | *Sal gre* | Shrub |
| | *Salvia regla* Cav. | *Sal reg* | Shrub |
| | *Scutellaria potosina* Brandegee | *Scu pot* | Herb |
| Liliaceae | *Echeandia flavescens* (Schult. & Schult. f.) Cruden | *Ech fla* | Herb |
| Linaceae | *Linum schiedeanum* Schltdl. & Cham. | *Lin sch* | Herb |
| | *Schoenocaulon texanum* Scheele | *Sch tex* | Herb |
| Nyctaginaceae | *Mirabilis oblongifolia* (A. Gray) Heimerl | *Mir obl* | Herb |
| Oleaceae | *Forestiera reticulata* Torr. | *For ret* | Shrub |
| | *Fraxinus greggii* A. Gray | *Fra gre* | Shrub |
| Onagraceae | *Calylophus berlandieri* Spach | *Cal ber* | Herb |
| | *Oenothera rosea* L'Hér. *ex* Aiton | *Oen ros* | Herb |
| Orchidaceae | *Goodyera oblongifolia* Raf. | *Goo obl* | Herb |
| | *Hexalectris grandiflora* (A. Rich. & Galeotti) L.O. Williams | *Hex gra* | Herb |
| | *Malaxis brachystachys* (Lindl.) Rchb. f. | *Mal bra* | Herb |
| Oxalidaceae | *Oxalis corniculata* L. | *Oxa cor* | Herb |
| | *Oxalis latifolia* Kunth | *Oxa lat* | Herb |
| Passifloraceae | *Passiflora suberosa* L. | *Pas sub* | Herb |
| Pinaceae | *Pinus cembroides* Zucc. | *Pin cem* | Tree |
| | *Pinus arizonica* Engelm. var. *stormiae* Martínez | *Pin ari* | Tree |
| Plantagiaceae | *Mecardonia vandellioides* (Kunth) Pennell | *Mer van* | Herb |
| Poaceae | *Achnatherum multinode* (Scribn. *ex* Beal) Valdés-Reyna & Barkworth | *Ach mul* | Herb |
| | *Bouteloua dactyloides* (Nutt.) Columbus | *Bou dac* | Herb |
| | *Bouteloua uniflora* Vasey | *Bou uni* | Herb |
| | *Brachypodium mexicanum* (Roem. & Schult.) Link | *Bra mex* | Herb |
| | *Bromus anomalus* Rupr. *ex* E. Fourn. | *Bro ano* | Herb |

| Family | Scientific Name | Species Abbreviations | Life Form |
|---|---|---|---|
| | *Bromus carinatus* Hook. & Arn. | *Bro car* | Herb |
| | *Elymus arizonicus* (Scribn. & J.G. Sm.) Gould | *Ely ari* | Herb |
| | *Muhlenbergia dubia* E. Fourn. | *Muh dub* | Herb |
| | *Muhlenbergia emersleyi* Vasey | *Muh eme* | Herb |
| | *Muhlenbergia glauca* (Nees) B.D. Jacks. | *Muh gla* | Herb |
| | *Muhlenbergia phleoides* (Kunth) Columbus | *Muh phl* | Herb |
| | *Muhlenbergia rigida* (Kunth) Kunth | *Muh rig* | Herb |
| | *Muhlenbergia setifolia* Vasey | *Muh set* | Herb |
| | *Nassella leucotricha* (Trin. & Rupr.) R.W. Pohl | *Nas leu* | Herb |
| | *Piptochaetium fimbriatum* (Kunth) Hitchc. | *Pip fim* | Herb |
| | *Schizachyrium sanguineum* (Retz.) Alston | *Sch san* | Herb |
| | *Trisetum filifolium* Scribn. *ex* Beal | *Tri fil* | Herb |
| | *Zuloagaea bulbosa* (Kunth) E. Bess | *Zul bul* | Herb |
| Polemoniaceae | *Loeselia greggii* S. Watson | *Loe gre* | Herb |
| Polygalaceae | *Hebecarpa barbeyana* (Chodat) J.R. Abbott | *Heb bar* | Herb |
| | *Polygala dolichocarpa* S.F. Blake | *Pol dol* | Herb |
| | *Polygala shinnersii* W.H. Lewis | *Pol shi* | Herb |
| | *Rhinotropis lindheimeri* (A. Gray) J.R. Abbott | *Rhi lin* | Herb |
| Polygonaceae | *Eriogonum atrorubens* Engelm. | *Eio atr* | Herb |
| Portulacaceae | *Talinum aurantiacum* Engelm. | *Tal aur* | Herb |
| Pteridaceae | *Pellaea intermedia* Mett. *ex* Kuhn | *Pel int* | Herb |
| | *Myriopteris rufa* Fée | *Myr ruf* | Herb |
| Ranunculaceae | *Clematis drummondii* Torr. & A. Gray | *Cle dru* | Herb |
| | *Clematis pitcheri* Torr. & A. Gray | *Cle pit* | Herb |
| Rhamnaceae | *Ceanothus greggii* A. Gray | *Cea gre* | Shrub |
| Rosaceae | *Lindleya mespiloides* Kunth | *Lin mes* | Shrub |
| | *Prunus cercocarpifolia* Villarreal | *Pru cer* | Shrub |
| | *Prunus serotina* Ehrh. | *Pru ser* | Shrub |
| | *Purshia plicata* (D. Don) Henrickson | *Pur pli* | Shrub |
| Rubiaceae | *Bouvardia ternifolia* (Cav.) Schltdl. | *Bou ter* | Shrub |
| | *Crusea diversifolia* (Kunth) W.R. Anderson | *Cru div* | Herb |
| | *Hedyotis palmeri* (A. Gray) W.H. Lewis | *Hed pal* | Herb |
| Santalaceae | *Phoradendron leucarpum* (Raf.) Reveal & M.C. Johnst. | *Pho leu* | Mistletoe |
| Scrophulariaceae | *Castilleja scorzonerifolia* Kunth | *Cas sco* | Herb |
| Solanaceae | *Nierembergia angustifolia* Kunth | *Nie ang* | Herb |
| | *Physalis hederifolia* A. Gray | *Phy hed* | Herb |
| | *Solanum verrucosum* Schltdl. | *Sol ver* | Herb |
| Verbenaceae | *Verbena neomexicana* Small | *Ver neo* | Herb |
| Violaceae | *Hybanthus verbenaceus* (Kunth) Loes. | *Hyb ver* | Herb |
| | *Hybanthus verticillatus* (Ortega) Baill. | *Hyb vrt* | Herb |

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
