# Peer review of "Livestock Grazing Impact on Species Composition and Richness Understory of the Pinus cembroides Zucc. Forest in Northeastern Mexico"

_forests, doi:10.3390/f13071113_

Round 1

Reviewer 1 Report

Line 24: Some studies have revealed……………….., better to remove this sentence and add your finding on species richness

Author Response

The suggestions of the referee has been implemented and can be checked in the track changes system of the MSWord Manuscript.

We thank the refereee for the comments.

Reviewer 2 Report

I found this paper to be interesting. However, I think the manuscript can be substantially improved. In general, I found the writing to be lacking. Some statements were vague to the point of having little to no meaning. Other statements were rather repetitive. The Materials and Methods and Results section is incomplete. Most importantly, it lacks a method of tree inventory. The results would also be better presented in a slightly different way. I find some results too detailed and others not presented in enough detail. The Discussion is somewhat under-developed and is partly a repetition of the results. It might be better to interpret the results in a slightly more detailed and less redundant way. My detailed comments are in the attached file and below.

L80-81: „In the protected area, pinyon pine forest occupies 12.54% of the area, and pinyon pine forest is associated with xeric scrubland 9.55%” - please rewrite this sentence, the second part is incomplete.

L114-115: Replace „densitometer” with „densiometer”.

L116-: What does „the same method” mean? What does „grass cover” exactly mean? Only cover of grasses or cover of all herbaceous plants?

L117: The method of tree layer inventory is missing, what data were recorded? I think, species and basal area, but you should describe exactly. Health status? Dead wood?

L173. The „low environmental variability” was expected, as this is how the sample was selected (all plots are on a relatively flat ridge).

L178. Is Table 1.b. required because the differences are small? I think we should have tables of other results, see below.

L180. Does this statement apply to the canopy layer or to all layyers? It seems to apply to the canopy, but it should be made clear.

L186. What do you mean by „average dominance percentage”? I think it would be useful to include tables of the density, basal area, and number of species in the canopy with a similar structure to Table 1.a. These would be more informative than Tables 1.a. or 1.b.

L192. Is 163 the total number of species in all layers? I think it only applies to the herb layer, but it should be made clear. It would be useful to include tables of the number of species in the this layer with a similar structure to Table 1.a.

L194. „In addition, we recorded 55 species in the exclusion plots, which did not occur in the grazed plots, while 34 species were present only in control plots.” I do ont understand – are not exclusion and control plots the same?

L216: Table 1a, 1b. It is a bit confusing that the averages are at the bottom and not below the treatment types.

L249-254: I do not necessarily agree with this statement, or cannot interpret it due to lack of data. Forest structure does not equal density and basal arrea. The distribution of size classes (diameter at breast hight – DBH – classes) is a better characteristic of the structure. Thus, for example, the density of large trees, and for the present research, the density and relative density of small size DBH classes is more important. Is there a difference in the diameter distribution between the two plot types? Has the volume of dead wood been measured?

L284: This is interesting, but it is just a list of species, does not it belong in a Results chapter?

Author Response

June 22th, 2022

forests-1763426

Dear Editor

            We carefully read the comments about the manuscript provided by the anonymous reviewers. We agree with much the comments, most of them have been fixed, and we have tried to clarify the remaining ones. We realized that the manuscript needed clarification in some parts, especially we have been pointing along the manuscript that the study is focusing on the understory species. We really appreciate the kindness of the referees and all their time dedicated to clearly specify all the changes. Also, some problems with the tables (they move in the final edition process) has been fixed.

The following specifies all the changes and replies to the comments:

Comments: I found this paper to be interesting. However, I think the manuscript can be substantially improved. In general, I found the writing to be lacking. Some statements were vague to the point of having little to no meaning. Other statements were rather repetitive. The Materials and Methods and Results section is incomplete. Most importantly, it lacks a method of tree inventory. The results would also be better presented in a slightly different way. I find some results too detailed and others not presented in enough detail. The Discussion is somewhat under-developed and is partly a repetition of the results. It might be better to interpret the results in a slightly more detailed and less redundant way. My detailed comments are in the attached file and below.

Answer: Base on these comments we try to clarify al the unclear aspect of the manuscripts. These will be recorded with the tracking system of MSWord. As said before, the paper hypothesis is the impact of grazing on understory species composition. Discussion is centered about the results on the understory species composition, comparing with results of similar experiments in other areas.

Comments: L80-81: „In the protected area, pinyon pine forest occupies 12.54% of the area, and pinyon pine forest is associated with xeric scrubland 9.55%” - please rewrite this sentence, the second part is incomplete.

Answer: We changed the sentence to: “In the protected area, pinyon pine forest occupies 12.54% of the area surrounded by a xeric scrubland (9.55% of the protected area)”

Comment: L114-115: Replace „densitometer” with „densiometer”.

Answer: Done

Comments: L116-: What does „the same method” mean? What does „grass cover” exactly mean? Only cover of grasses or cover of all herbaceous plants?

Answer: We change the beginning of that sentences. We include (only grasses, excluding forbs). We were interested in focusing only on grasses cover in this case as an important factor affected by grazing.

Comments: L117: The method of tree layer inventory is missing; what data were recorded? I think, species and basal area, but you should describe exactly. Health status? Dead wood?

Answer: We included: We measure DBH for all trees alive in the 30x30 m plots to estimate basal area and density to ha. (we only took this information as long as we were interested in the forest structure).

Comment: L173. The „low environmental variability” was expected, as this is how the sample was selected (all plots are on a relatively flat ridge).

Answer: this part was removed.

Comment: L178. Is Table 1.b. required because the differences are small? I think we should have tables of other results, see below.

Answer: We consider the table 1b is necessary to provided the information of the lack of nutrient differences, as this was one of the hypotheses of this study.

Comment: L180. Does this statement apply to the canopy layer or to all layers? It seems to apply to the canopy, but it should be made clear.

Answer: We have changed this sentence to: Altitude of the plots ranged between 2350-2500 and as for slope and canopy cover, differences were not significant for control vs. grazed plots.

L186. What do you mean by „average dominance percentage”? I think it would be useful to include tables of the density, basal area, and number of species in the canopy with a similar structure to Table 1.a. These would be more informative than Tables 1.a. or 1.b.

 Answer: We agree that that information is of interest, however, the main hypothesis of the study is focusing on the understory species composition, not canopy composition, as long as 20 years will not provided differences in that way. Because of that we provided the basal area, density and species richness to provide the structure information.

We change the sentence: Pinus cembroides was the dominant species with higher values of basal area in grazed plots (85.6%) than in control plots (72.0%).

Comments: L192. Is 163 the total number of species in all layers? I think it only applies to the herb layer, but it should be made clear. It would be useful to include tables of the number of species in the this layer with a similar structure to Table 1.a.

Answer: We included the word understory to clarify it. Also, the richness and evenness was graphically represented. The study focus on the understory and we considered that a graph will be more informative.

Comments: L194. „In addition, we recorded 55 species in the exclusion plots, which did not occur in the grazed plots, while 34 species were present only in control plots.” I do ont understand – are not exclusion and control plots the same?

Answer: Right, we have corrected these two sentences now.

Comments: L216: Table 1a, 1b. It is a bit confusing that the averages are at the bottom and not below the treatment types.

Answer: We have done the suggested changes.

L249-254: I do not necessarily agree with this statement, or cannot interpret it due to lack of data. Forest structure does not equal density and basal arrea. The distribution of size classes (diameter at breast hight – DBH – classes) is a better characteristic of the structure. Thus, for example, the density of large trees, and for the present research, the density and relative density of small size DBH classes is more important. Is there a difference in the diameter distribution between the two plot types? Has the volume of dead wood been measured?

Answer: We agree, but as long as we are centering the study on species understory, we consider that all that suggestions will a different study. In order to center more the manuscript we change also the title including the word “understory”. But our main hypothesis is that understory is changing base in management, and that structure base in basal area and density does not differ, in order to isolate the impact of grazing.

 L284: This is interesting, but it is just a list of species, does not it belong in a Results chapter?

Answer:The study is focusing on the understory plant community, the results of diversity, richness and evenness are one the objectives to analyze and to compare with other studies. We have described the information on results, but on discussion we are comparing our results trying to create a study framework with respect other similar studies.

Once again, thanks to the referee for the work and time dedicate for the improvement of the manuscript.

Reviewer 3 Report

L33:…, as these cases can cause

L35:…overgrazing by cattle, horses

L66-67: You may combine the two sentences. …, and belongs to the…

L167: An MRPP (….) was used …

Figure 2: You may label the bars the way you describe them in the caption (or visa versa).

L249: …, and resource availability.

L264: You may omit “the.” (…explained species composition…)

L305: Do you mean a reduction in the total number of trees and number of species?

L309: You may omit the comma in front of “should.”

L310: You may omit the comma.

Author Response

Comments: L33:…, as these cases can cause

L35:…overgrazing by cattle, horses

L66-67: You may combine the two sentences. …, and belongs to the…

L167: An MRPP (….) was used …

 Figure 2: You may label the bars the way you describe them in the caption (or visa versa).

 L249: …, and resource availability.

L264: You may omit “the.” (…explained species composition…)

L305: Do you mean a reduction in the total number of trees and number of species?

L309: You may omit the comma in front of “should.”

L310: You may omit the comma.

Answer: All the suggestions were implemented. These can be checked with the trackchanges systems in the MSWord file submitted.

We thank the referee for the time dedicated to the improvement of the manuscripts.

Round 2

Reviewer 2 Report

In my opinion the manuscript has been improved. I just have fewer comments remaining. Layers are still not clear everywhere. I understand that the manuscript focuses on the impact of grazing on understory species composition, but the depth of the discussion of the canopy layer is not always clear. I think the Discussion needs to be expanded.

My detailed comments are:

1. Comment: L180 (L185 in revised version). Does this statement apply to the canopy layer or to all layers? It seems to apply to the canopy, but it should be made clear.

This comment was not answered. So, to which layer does this statement apply? If you are referring to canopy level, I see a contradiction between the objectives, the answer („the study is focusing on the understory species”) and the text. Whether the paragraph refers to the canopy or the understorey layer, the text should be clarified.

2. Comments and answer L192. “The study focus on the understory and we considered that a graph will be more informative.”

You are right, but so the mean and standard deviation can only be approximated, not at all for individual plots, while for the much less informative soil and abiotic information data this is shown in the table. I would consider some more detailed data presentation.

3. The Discussion could be extended to include an evaluation of different groups of species between the two types of treatment. For example, such groups could be native / adventive or generalist / specialist species.

Author Response

In my opinion the manuscript has been improved. I just have fewer comments remaining. Layers are still not clear everywhere. I understand that the manuscript focuses on the impact of grazing on understory species composition, but the depth of the discussion of the canopy layer is not always clear. I think the Discussion needs to be expanded.

Respond: We thank the referee for his time and dedication to improve the manuscript and we have follow his indications. We have expanded some of the comments on the discussion now.

My detailed comments are:

  1. Comment: L180 (L185 in revised version). Does this statement apply to the canopy This comment was not answered. So, to which layer does this statement apply? If you are referring to canopy level, I see a contradiction between the objectives, the answer („the study is focusing on the understory species”) and the text. Whether the paragraph refers to the canopy or the understorey layer, the text should be clarified.

Respond: We describe the category of “tree” on the discussion, in the sampling design section. In order to clarify it, we added now at the end of the first paragraph of “Sampling Design” the following sentence: This category will be considered as part of the canopy for its description.

  1. Comments and answer L192. “The study focus on the understory and we considered that a graph will be more informative.”

You are right, but so the mean and standard deviation can only be approximated, not at all for individual plots, while for the much less informative soil and abiotic information data this is shown in the table. I would consider some more detailed data presentation.

Respond: We incorporated the information of the canopy from a canopy description point of view. The main objective of this study is the understory plant community. Other study has been published centered just on the canopy. We referred to it in the text now, so the detailed information of the canopy can be found there.

  1. The Discussion could be extended to include an evaluation of different groups of species between the two types of treatment. For example, such groups could be native / adventive or generalist / specialist species.

Respond: We extended our comments about the species in relation with the biogeographical origin, although only one species was found as exotic (grazed plots). It means that environmental filters make difficult for exotics to get established there. We also include an additional refence.